# Patients and healthcare professionals perspectives on creating a chronic pain support line in Portugal: A qualitative study protocol

**Mariana Cruz**[1][*], **Simão Pinho**[1], **José Manuel Castro-Lopes**[1,2], **Rute Sampaio**[1,3]

1 Department of Biomedicine, Faculty of Medicine of University of Porto, Porto, Portugal, 2 i3S Institute for Research & Innovation in Health, Porto, Portugal, 3 CINTESIS—Center for Health Technology and Services Research, Porto, Portugal

☯ These authors contributed equally to this work.
* m.belo.cruz@gmail.com

## Abstract

Chronic pain affects almost 38% of the Portuguese adult population, with high costs for both patients and society. Those who suffer with chronic pain frequently complain of feeling misunderstood and of lack of support. These complaints are the main reason why support telephone lines for chronic pain were created in some countries. However, there is no scientific data supporting their creation or evaluating their performance. This paper presents a qualitative study protocol to assess patients and healthcare professionals' perspectives on the creation of a telephone support line for chronic pain. It constitutes the first step to attain the main goal of developing and implementing a functioning support line for chronic pain in Portugal. The methodology to assess patients and healthcare professionals' perspectives and needs is presented. In order to gather information as close to reality as possible, focus groups interviews were chosen as data sources. Given the present context of the COVID-19 pandemic, meetings will take place online, using a digital platform. All interviews will be transcribed verbatim, coded and synthesised into categories and main themes. Thematic analysis will be conducted using NVivo® V12 software, employing an iterative and reflexive approach. Finally, comparative and relational analysis will be performed in order to identify topics where patients and professionals converge or greatly diverge. The findings will be useful for grounding the creation of a telephone support line for chronic pain patients. Results dissemination will be important for policy-makers to develop a new perspective towards chronic pain services available.

## Introduction

According to the International Association for the Study of Pain (IASP), pain is "an unpleasant sensory and emotional experience associated with, or resembling that associated with, actual or potential tissue damage" and a major cause of suffering and disability in the world [1, 2]. As

**Data Availability Statement:** No datasets were generated or analysed during the current study. All relevant data from this study will be made available upon study completion.

**Funding:** The authors received no specific funding for this work.

**Competing interests:** The authors have declared that no competing interests exist.

for chronic pain, it is defined as pain that persists or recurs for longer than 3 months and is associated with significant emotional distress or functional disability [1]. In 2001, the European Federation of Pain (EFIC) recognised chronic pain as a pathology in itself, whether isolated or accompanied by other disorders [3]. It should be noted that chronic pain can be divided into primary chronic pain (without an obvious cause), or secondary chronic pain (as a symptom of an underlying pathology, including cancer, post-surgical, post-traumatic, neuropathic, orofacial, visceral and musculoskeletal pain) [1].

Recent studies revealed that chronic pain prevalence in European adult population is of 20%, with similar numbers being found in the US [4, 5]. Among the adult Portuguese population, prevalence exceeds 30%, making it a major public health problem [6]. The consequences are not only personal but also societal, such as the costs of absenteeism, early retirement and social benefits. These aspects contribute to total annual costs of 5,635.26 million euros per year in Portugal. Despite these numbers, the same study concluded that only 1% of Portuguese chronic pain patients has access to specialized follow-up [7].

There is a particular duality to chronic pain in Portugal: access to care is somewhat restricted (given the difficulties in having access to a family physician, it is even more difficult to access chronic pain specialists), but when someone uses health services because of their pain, there is a tendency for over-using it, with no clinical gain [7, 8]. For this reason, there is a need for interventions that bring patients and healthcare professionals closer, in order to empower those who suffer and maximize the effective use of the available resources [7]. The maximization of services includes avoiding its unjustified use (providing patients with tools for pain prevention and management) and tackling major issues, such as treatment adherence (taking the correct dosage, of the correct medicine, on a timely manner and during the prescribed time of treatment).

The information described so far partly justifies the creation of a chronic pain support telephone line, as it will facilitate access to healthcare professionals, providing empathic listening, patient rehabilitation, empowerment and appropriate referral, while also making it possible to implement adherence enhancing interventions. There are some pain support lines already functioning in some countries, namely in Ireland, Canada, England and Australia [9–12]. Most of these lines are volunteer-operated and were created by private or aid organizations. Consequently, they are not submitted to performance evaluation. Nevertheless, recent experiments with telecare for chronic pain patients show positive impact regarding disease management, and users' testimonies reinforce the importance of human presence for the patient suffering with chronic pain [13].

Considering the high prevalence of chronic pain in Portugal, it would be important to have such a support service for those living with the disease. A telephone support line would be available for patients living in both metropolitan and rural areas, whether they are housebound or not. This would make healthcare access more equal, extending support to anyone who needs it and has access to a telephone, with a relatively widespread, inexpensive and accessible technology. Simultaneously, it would make possible to tackle major issues such as healthcare services overuse and medication adherence, contributing to an important reduction of costs [13, 14].

Collecting the opinions and perspectives of patients with chronic pain and healthcare professionals working in the area is essential, as this is the only way to provide a truly objective ground for the creation of a support line. This information will allow the development of a project based on the reality of those who live with the disease on a daily basis, a foundation that will permit, in the long term, a precise and objective evaluation of the results and impact of the support line.

In order to gather such information, a direct interaction with chronic pain patients and healthcare providers is needed, preferably in a group setting. Being so, the decision for focus group interviews was made, for this method allows to obtain a diverse array of perspectives, as close to everyday reality as possible.

To our knowledge, this is the first qualitative study that aims to objectively validate the creation of a support line, as well as gathering information that is essential to the development of a service adapted for those who will benefit from it [13, 14].

## Materials and methods

### Aims and objectives

The study will be conducted under the scope of the project "Creating a pain support line in Portugal: feasibility, development and impact". Its main objective is the creation of a telephone support line for chronic pain patients. More specific objectives are: 1) evaluating chronic pain patients and healthcare professionals needs and expectations (including what kind of services are expected), 2) developing a decisional algorithm that provides adequate advice for each situation, 3) providing healthcare professionals that will operate the line with specific training on chronic pain, 4) supplying confidential support to those in need while assuring communication between multiple agents in healthcare services and 5) warranting flexibility and respect for patient and healthcare professionals individuality. This study protocol will focus mainly on the first objective, describing the qualitative study designed to fulfil it.

### Study design and setting

In order to evaluate chronic pain patients and healthcare professionals' needs and expectations, an exploratory and phenomenological qualitative design study will be performed. As such, different qualitative methods were considered to determine the best data collection procedure, such as semi-structured interviews, focus groups, questionnaires, among others. After careful consideration, the research team opted in favour of direct interaction with chronic pain patients and healthcare providers, in line with the notion that human interaction plays an essential part in chronic pain management. By interacting with subjects in a controlled group setting, that is, performing focus group interviews, one can obtain a diverse array of perspectives, as close to everyday reality as possible. This is particularly effective because group discussions with peers provide participants with a safe and familiar environment to express their opinions and describe their experiences [15]. Ideally, the focus groups would consist of an in-person gathering, which would allow information other than spoken words, such as body language and posture for example, to be registered. However, due to COVID-19 pandemic restraints, the focus groups will take place via online videoconference, using the "Zoom" platform. The recommendations for online focus-groups are that the number of participants do not exceed five to six, because a fluid discussion is difficult to obtain otherwise [16, 17]. Thus, we aim to have minimum of 1 group for both patients and healthcare professionals, composed of 5 to 6 participants each.

After being invited to participate, a presentation regarding focus groups dynamics and videoconference characteristics and requirements will be held for both patients and healthcare professionals who volunteer to participate, before the meetings themselves have place. They will also be informed that all meetings will be recorded and transcribed verbatim, always in accordance with the Portuguese data protection laws.

Qualitative data will be analysed with the NVivo® V12 software to provide evidence to guide the creation of the support line. The next step of this project consists of evaluating the feasibility of creating a decisional algorithm, through collaboration with a panel of researchers with competencies both on the field of chronic pain and on informatics, particularly on

decisional algorithms. The algorithm will then be subjected to all the necessary corrections and adjustments, which will be possible with the creation of a test-line to be assessed with the collaboration of patients and healthcare professionals. The pivot-line will then be amplified, according to stakeholders' decisions and resources available.

## Participants

The study will be presented by the research team to an Association of People with Pain (Força-3P) and to the Portuguese Association for the Study of Pain (APED) members, who will serve as a connection between the research team and those who volunteer to participate in the study. The following characteristics will be considered as inclusion criteria of participants: being 18 years or older, having the ability to understand and communicate in Portuguese, being able to use technologies and log in a Zoom call, and being a chronic pain patient (diagnosed at least 2 years ago) or a healthcare professional working with chronic pain patients.

**Healthcare professionals.**   Healthcare professionals (e.g., medical doctors, nurses, psychologists) will be volunteers from different areas of healthcare that deal with chronic pain, such as palliative care nurses and oncologists, for example. After attending the presenting session, a date will be arranged for the focus group. Written consent will be obtained prior to the meeting.

**Chronic pain patients.**   Patients will be recruited by the representatives of the National Association of People with Chronic Pain (Força-3P), who will reiterate that participation is voluntary. In case of agreement, the representatives will send an e-mail to the patient, inviting them to the presentation session. Then, the representatives will help establishing communication between the patients and research team, in order to arrange meeting details. Written consent will be obtained beforehand.

## Data collection procedures

The online focus group sessions are set to be synchronous and to last from 60 to 90 minutes [17]. An entirely online approach was chosen to ensure safety of participants in times of pandemic. The participants will be free to choose to use a codename, so that they may remain anonymous. Each group will be led by two moderators with backgrounds in research and work on chronic pain. They will be present 30 minutes before the starting time to solve any technical issues. The principal investigator will act as a group facilitator and guide the interview, following a semi-structured focus group guide designed to elicit participants' experiences. The second moderator will be responsible for recording major quotes, non-verbal interactions, and expressions, in order to add context to the recordings and aid discussion of the transcripts later on. Each focus group session will include an overview to introduce the objective, a series of questions which proceed from general to specific, and a summary to highlight and verify key points discussed (**Tables 1** and **2**) [15].

The major topics covered in the focus groups are based on literature regarding chronic pain patients and healthcare professionals' experiences, as well as on topics related to technology potential and applications [18–24].

This approach will enable the researchers to collect information about the patients' and healthcare professionals' perceptions regarding the services currently available, needs that remain to be met and expectations for a chronic pain support line.

## Data analysis

All focus group data will be transcribed verbatim and anonymized. After that, thematic and content analysis will be performed through an iterative and reflexive process, carried out by

**Table 1. Patients' focus group theme guide.**

| Major themes | Specific discussion themes |
|---|---|
| Perspectives on the existing and/or lack of support for patients | · Reaching out for support |
| | · Types of support needed |
| Perspectives on accessibility of such support methods | · Support accessibility |
| | · Enhancing accessibility for all |
| Perspectives on pain medication adherence | · Effectiveness of pain medication |
| | · Importance of pain medication to pain management |
| | · Needs for information regarding pain medication |
| Perspectives on using information technologies as a mean of support | · Perspectives on using information technologies (such as mobile-phones, apps, e-mails, etc) as support methods |
| | · Utility of these vehicles of support |
| | · Usage of these support methods when needed |
| Perspectives on the creation of a telephone line | · Perspectives on the creation of a support telephone line |
| | · Needs expected to see fulfilled |
| | · Reaching to this kind of support when trying to manage pain |
| | · Applicability of such line in the day-to-day life of patients |

two researchers working independently. Next, the transcripts will be imported into the NVivo®12 software and codification will take place [25]. Coding of the segments of the transcriptions will be made quotation by quotation [26]. The segments of coded text will be synthesised into categories, which will be further grouped into main themes [27, 28]. Findings will emerge directly from raw data, firstly using an inductive coding technique and, after that, applying the deductive approach to the same transcript. Each technique will be carried out independently for the patients and for healthcare professionals focus groups, in a systematic, sequential, and continuous way. The use of two coding techniques is expected to reduce possible bias of the codifiers.

Analysis of the obtained data will be made according to the framework method. In a first stage, each focus group will be analysed separately, through a detailed description and

**Table 2. Healthcare professionals' focus group theme guide.**

| Major themes | Specific discussion themes |
|---|---|
| Perspectives on the existing and/or lack of support for patients | · Effectiveness of existing support |
| | · Support services available in the healthcare system |
| | · Knowledge of chronic pain patients' opinions regarding available support |
| Perspectives on accessibility of such support methods | · Support accessibility |
| | · Enhancing accessibility for all |
| Perspectives on pain medication adherence | · Importance of pain medication adherence |
| | · The real medication adherence of patients |
| | · Usefulness of a service potentially providing information regarding pain medication |
| Perspectives on using information technologies as a mean of support | · Perspectives on using information technologies (such as mobile-phones, apps, e-mails, etc) as support methods |
| | · Usefulness of these vehicles of support |
| | · Real-life usefulness of these support methods for patients |
| Perspectives on the creation of a telephone line | · Perspectives on the creation of a support telephone line for patients |
| | · Needs expected to see fulfilled |
| | · Integrating this kind of support in the approach to patients' care |

interpretation of main themes, comparing the results of both coding techniques. In the next step, comparative and relational analysis from both focus groups will be carried out, in order to identify how the perspectives of chronic pain patients and healthcare practitioners may converge or diverge in specific topics.

All data will be discussed and interpreted by 3 researchers, the 2 codifiers and a third researcher to provide another perspective and help solve disagreements. Eventual discrepancies in codification will be discussed until consensus is reached. All findings will be reported following the COREQ guideline (Consolidated Criteria for Reporting Qualitative Research) [29].

## Data management

Data retrieved from focus groups will be recorded, anonymised, analysed and published. It will be preserved in private folder, protected with password, in a secure computer, for the researchers' use and validation purposes only. There is no intention to reuse this data; thus, it will be stored until conclusion of the main project and destroyed afterwards.

## Ethical considerations

The protocol of the project "Creating a pain support line in Portugal: feasibility, development and impact.", in which this study is integrated, was approved by the Ethics Committee of the Centro Hospitalar Universitário de São João, Faculty of Medicine of University of Porto (109–21). An information sheet with a description of the study and its objectives will be given to all participants, providing them ways to contact the research team. Each participant will be given a written informed consent for the interview recordings and collected materials. All data will be anonymous and confidential. Data protection will be ensured by separating audio records, transcripts, consents and questionnaires, as well as by using passwords to access them. Any code linking data to individuals will be safely stored and only accessible to the research team. All participants will be anonymous throughout the entire study and its dissemination, from a data interpretation perspective.

## Discussion

To our knowledge, this will be the first study to gather scientifically grounded data to support the development of a chronic pain support line, through patients and healthcare professionals' perspectives. This knowledge will also enable us to objectively evaluate the impact that the line may have. Rigorous standards of qualitative research, such as dependability, credibility, confirmability and transferability will ensure that the findings obtained are consistent with the methods of the interpretivist paradigm and its information sources [27, 30].

The study has some anticipated limitations that should be mentioned. A potential limitation is the possibility of over-representation of groups that are more likely to participate in a time-consuming activity like the focus groups, such as unemployed people or people on sick leave. Likewise, convenience sampling is not an ideal sampling method, and we can foresee that our sample will not an ideal representation of the whole national population with chronic pain. Given that the organizations of contact are seeded in an urban environment, it is expected there to be some underrepresentation of rural population, for example. These limitations are, at least during this time of pandemic, when access to healthcare services is restricted to those who need them, difficult to overcome. However, the gathered data will still be very useful, and a significant first step in the qualitative approach to a better understanding of the needs and expectations of Portuguese chronic pain patients and healthcare practitioners. Moreover, it will also provide new insights on how a support telephone line might attend to the identified

needs, while allowing for a better use of healthcare services. The added value of this study lies on its concern with underlying perceptions, needs and attitudes towards the use of technology as a means of bringing support closer to and helping those who suffer with chronic pain. Finally, the combined perspectives of both potential users (chronic pain sufferers whom might benefit from the support line) and healthcare professionals, ensures that all relevant views are taken into account.

This qualitative study will allow us to gather evidence on the real-life chronic pain context and ensure that every step of the project will be linked to the real needs of those to whom the support line is intended. Moreover, this protocol could constitute a useful tool to encourage other researchers towards the development of a chronic pain support line, as it fosters the application of a widespread and accessible technology.

Results will be published in international peer-reviewed journals and presented in conferences. Furthermore, the end results will be disseminated nationally in seminars directed to the general public and policy-makers in the health and social sectors.

## Acknowledgments

We are most grateful to the Força-3P Association, (National Association of Chronic Pain Patients) who agreed to participate in the study, for their support, time and patience. We are also deeply grateful to all healthcare professionals who agreed to contribute with their knowledge and experience in working with chronic pain patients.

## Author Contributions

**Conceptualization:** Mariana Cruz, Simão Pinho, José Manuel Castro-Lopes, Rute Sampaio.

**Methodology:** Mariana Cruz.

**Supervision:** José Manuel Castro-Lopes, Rute Sampaio.

**Writing – original draft:** Mariana Cruz, Rute Sampaio.

**Writing – review & editing:** Simão Pinho, José Manuel Castro-Lopes, Rute Sampaio.

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
