## [Decision Letter · Decision Letter 0]

26 May 2022

PONE-D-21-35572Patients and healthcare professionals perspectives on creating a chronic pain support line in Portugal: a qualitative study protocolPLOS ONE

Dear Dr. Cruz,

Thank you for submitting your manuscript to PLOS ONE. After careful consideration, we feel that it has merit but does not fully meet PLOS ONE’s publication criteria as it currently stands. Therefore, we invite you to submit a revised version of the manuscript that addresses the points raised during the review process. The reviewers have highlighted important aspects to improve this study protocol. Specifically, although there cannot be results (being it a protocol), all methodological aspects needs to be clarified so that this paper can inspire close replication and close/similar research in the field. 

We look forward to receiving your revised manuscript.

Kind regards,

Sara Rubinelli

Academic Editor

PLOS ONE

Journal Requirements:

Reviewers' comments:

Reviewer's Responses to Questions

**Comments to the Author**

1. Does the manuscript provide a valid rationale for the proposed study, with clearly identified and justified research questions?

Reviewer #1: Yes

Reviewer #2: Partly

Reviewer #3: Yes

2. Is the protocol technically sound and planned in a manner that will lead to a meaningful outcome and allow testing the stated hypotheses?

Reviewer #1: Yes

Reviewer #2: No

Reviewer #3: Partly

3. Is the methodology feasible and described in sufficient detail to allow the work to be replicable?

Reviewer #1: Yes

Reviewer #2: No

Reviewer #3: No

4. Have the authors described where all data underlying the findings will be made available when the study is complete?

Reviewer #1: Yes

Reviewer #2: No

Reviewer #3: Yes

5. Is the manuscript presented in an intelligible fashion and written in standard English?

Reviewer #1: Yes

Reviewer #2: Yes

Reviewer #3: Yes

6. Review Comments to the Author

You may also provide optional suggestions and comments to authors that they might find helpful in planning their study.

Reviewer #1: Thanks for the Authors for the valuable protocol that looks in an issue which require a good care. looking forwards to see the study manuscript and its results.

Reviewer #2: The manuscript written by Cruz et al describes the procedure implemented by the researchers in order to gather patients’ and healthcare’s professional standpoints about the Implementation of a support line for chronic pain patients in Portugal.

I have some concerns about the information provided by the authors that merit further consideration:

In my opinion, the main problem of this manuscript that there are major gaps in the information provided in terms of the preparation of the scientific protocol and the presentation of the results. The latter are reduced to few paragraphs in the discussion without sharing the real numbers obtained from the study which makes impossible to elaborate an exhaustive review of the present manuscript.

Even though authors make clear the need for such support line and the potential benefit for the Portuguese population suffering of chronic pain, there are some gaps in the elaboration of the context:

- In line 58, the authors cited “recent studies” that revealed the prevalence of chronic pain in Europe and the US. However, those references dated from 2016 and 2013, which makes me wonder if there are not more recent studies that could be more accurate with the present situation, moreover, taking into consideration the advances in pain management that have occurred in the last year.

- In line 64, the authors mentioned that access to care is “somehow restricted” but it is not specified why or where. Are they talking in general in the developed countries? Only in Portugal? Why access to care is restricted and for whom? As this is one of the central pillars to justify the need of implementing the support line, it should be much more elaborated and rationalized.

- In line 76, it is mentioned that there are currently similar systems already implemented in different countries. Please add here references or links to those systems to help both the reviewers and the readers to get an idea. Moreover, authors explain how those lines previously referred, are having a positive impact on user’s lines. However, the reference alludes to a study performed in the US (not even mentioned among the countries listed before).

- Taking into consideration the very well know differences between the healthcare systems across countries and how different the access to basic or specialized care could be depending on having public, private or semiprivate health infrastructures, I would appreciate that this issue would have been considering when presenting the effect of a support line. Indeed, the explanation for the expected benefit of such implementation in Portugal should have been based on the results obtained in countries with similar systems.

Concerning the Study design, I would appreciate to see a clearer and more detailed timeline of the implementation of the study, including a more exhaustive description of the study groups (number of groups, number of people participating, distribution of participants according to their profiles, gender, age). How were participants distributed in the groups? Has any participant been rejected?

Even if Tables 1 and 2 collect the Major and Specific discussion items, it remains not clear how those aspects were addressed during the meetings.

Nowadays, data privacy is one of the major topics when refers to research involving human beings, even if only healthy subjects are recruited. In that direction, authors have not provided any information concerning where the videos from the groups were stored, how they were protected from being accessed by non-authorized staff, how privacy has been secured…

However, as previously mentioned, the main problem of this manuscript is that the authors do not present any results that can be reviewed or contrasted. They mentioned in line 232 a potential limitation due to an over-representation of groups that the reviewers cannot judge because there is no data provided to support any discussion or conclusion.

Reviewer #3: The authors have presented a nice paper about the protocol adopted to evaluate and validate the possibility to set up "a chronic pain support line" in Portugal.

The topic is important to be discussed and the unmet need of this approach is well described by the authors.

Unfortunately there are some major concerns that should be addressed before considering it for possibile publication

The major concern is a methodological issue that should be better clarified. It is not clear how you have defined the criteria to "balance" chronic non oncological and oncological patients and healthcare providers. This is really important as the two different populations of patients and also of caregiver providers could have different needs.

Another concern is always related to the selection of your population who will evaluate the "support line". In the text it is not described if the authors have taken in consideration how to avoid bias related to select people only from countryside/city, type of pain, etc etc. Please describe better which are all the parameters that you consider in the evolution of patients and healthcare providers selected. If you have not planned it please describe it better in limitations of the protocol

The authors should describe better the protocol in order to permit the possibility to replicate in another country if possible.

7. PLOS authors have the option to publish the peer review history of their article (what does this mean?). If published, this will include your full peer review and any attached files.

Reviewer #1: **Yes: **Hossam ELDien A. ELShamaa

Reviewer #2: No

Reviewer #3: **Yes: **Massimo Allegri

---

## [Author Response · Author response to Decision Letter 0]

8 Jul 2022

Editor’s Comments:

"The reviewers have highlighted important aspects to improve this study protocol. Specifically, although there cannot be results (being it a protocol), all methodological aspects needs to be clarified so that this paper can inspire close replication and close/similar research in the field."

-We thank the reviewers for their valuable insight, and have further detailed the study’s methodology. Please find the appropriate changes in the updated document.

"Please ensure that your manuscript meets PLOS ONE's style requirements, including those for file naming. (…)"

-The manuscript’s formatting has been revised, and we believe it is compliant with PLOS ONE’s requirements.

"In your Data Availability statement, you have not specified where the minimal data set underlying the results described in your manuscript can be found. PLOS defines a study's minimal data set as the underlying data used to reach the conclusions drawn in the manuscript and any additional data required to replicate the reported study findings in their entirety. All PLOS journals require that the minimal data set be made fully available."

-This study has no results description as it is a protocol. In the Data Availability Statement, we will make all data used to reach the work’s conclusions when we have results (“All relevant data from this study will be made available upon study completion.)

"Upon re-submitting your revised manuscript, please upload your study’s minimal underlying data set as either Supporting Information files or to a stable, public repository and include the relevant URLs, DOIs, or accession numbers within your revised cover letter."

-Since our work is a study protocol and there are no results, all the information available and relevant has been detailed in the manuscript.

Your ethics statement should only appear in the Methods section of your manuscript. If your ethics statement is written in any section besides the Methods, please delete it from any other section. 

-This issue has been rectified as instructed.

Reviewers' Comments:

Reviewer #1:

“Thanks for the Authors for the valuable protocol that looks in an issue which require a good care. looking forwards to see the study manuscript and its results.”

-We would like to thank reviewer #1 for their positive reception of our manuscript; it encourages us to keep working on this project.

Reviewer #2: 

(…) “there are major gaps in the information provided in terms of the preparation of the scientific protocol and the presentation of the results. The latter are reduced to few paragraphs in the discussion without sharing the real numbers obtained from the study which makes impossible to elaborate an exhaustive review of the present manuscript.”

-We thank reviewer #2 for their remark. Since this is a study protocol, there are no results to be reported in this manuscript. Nevertheless, we have strived to be as transparent as possible and predict the project’s evolution as best as one can without incurring in biases. As such, it is specified that the results will be reported according to the COREQ guideline, ensuring the scientific soundness of our work (Please see the Data Analysis subsection in the Materials and Methods section, particularly the last paragraph). 

“Even though authors make clear the need for such support line and the potential benefit for the Portuguese population suffering of chronic pain, there are some gaps in the elaboration of the context:

- In line 58, the authors cited “recent studies” that revealed the prevalence of chronic pain in Europe and the US. However, those references dated from 2016 and 2013, which makes me wonder if there are not more recent studies that could be more accurate with the present situation, moreover, taking into consideration the advances in pain management that have occurred in the last year.”

-We agree with reviewer #2’s assertion that it is important to include recent studies in order to strengthen our contextualization. In fact, we did include references to two studies that were written in 2013 and 2016 (references 4. and 7.), because we believe they are still very effective in providing context to the current impact of chronic pain; however, it is also referrenced on the same line (line 58) a study published in 2021 on Pain (reference 5.). Furthermore, other recent studies are focused on specific populations and not chronic pain patients in general. Finally, even though we recognise the importance of recent advances in pain management, we believe this is a different topic, which does not pertain to what is being discussed in the mentioned manuscript excerpt, i.e. chronic pain prevalence.

“- In line 64, the authors mentioned that access to care is “somehow restricted” but it is not specified why or where. Are they talking in general in the developed countries? Only in Portugal? Why access to care is restricted and for whom? As this is one of the central pillars to justify the need of implementing the support line, it should be much more elaborated and rationalized.”

-We thank reviewer #2’s constructive remark. We have changed the wording on that sentence, by adding more detail and, thus, making the meaning clearer.

“- In line 76, it is mentioned that there are currently similar systems already implemented in different countries. Please add here references or links to those systems to help both the reviewers and the readers to get an idea. Moreover, authors explain how those lines previously referred, are having a positive impact on user’s lines. However, the reference alludes to a study performed in the US (not even mentioned among the countries listed before).”

-We very much appreciate reviewer #2’s insight. The mentioned excerpt was indeed quite confusing and somewhat incomplete. We have rectified this in the updated version.

“- Taking into consideration the very well know differences between the healthcare systems across countries and how different the access to basic or specialized care could be depending on having public, private or semiprivate health infrastructures, I would appreciate that this issue would have been considering when presenting the effect of a support line. Indeed, the explanation for the expected benefit of such implementation in Portugal should have been based on the results obtained in countries with similar systems.”

-We find reviewer #2’s comment very pertinent and thank them for their insight. In fact, unfortunately it is impossible to reliably predict the impact of a chronic pain help line in Portugal by comparing it to other lines in countries with similar healthcare systems: chronic pain help lines are rare worldwide, and the examples that exist are pro bono initiatives that are not audited and whose efficacy and impact are not evaluated in any meaningful way. This is one of the reasons we believe our project is so promising and innovative: beyond being a chronic pain help line, it is evidence-based, well-founded in scientific research and taking into account various perspectives from stakeholders. Furthermore, a quality control mechanism will be implemented, and the line’s effectiveness and economic impact will be evaluated.

“Concerning the Study design, I would appreciate to see a clearer and more detailed timeline of the implementation of the study, including a more exhaustive description of the study groups (number of groups, number of people participating, distribution of participants according to their profiles, gender, age). How were participants distributed in the groups? Has any participant been rejected?”

-We understand reviewer #2’s concern, however, since this work is a protocol and there are no participants yet, we cannot provide their description. Nevertheless, we have added more information regarding the sample of patients we expect to gather (please read the final sentence of the first paragraph of the Study design and setting section. 

“Even if Tables 1 and 2 collect the Major and Specific discussion items, it remains not clear how those aspects were addressed during the meetings.”

-In Tables 1 and 2, the theme guides for both discussion groups are addressed. We understand the reviewer’s request, but, since this work is a protocol, we cannot provide details about how themes were addressed because the meetings have not yet taken place.

“Nowadays, data privacy is one of the major topics when refers to research involving human beings, even if only healthy subjects are recruited. In that direction, authors have not provided any information concerning where the videos from the groups were stored, how they were protected from being accessed by non-authorized staff, how privacy has been secured…”

-We thank reviewer #2’s reminder of the importance of data privacy and security. In our study, as described in this protocol, we will take all necessary precautions to ensure patient data is handled correctly. Moreover, this manuscript strictly complies with all data protection laws in Portugal and the EU. Firstly, as described in detail, all patients will sign consent forms and have the choice to choose an alias and keep their camera turned off during the focus group session, so that they may remain anonymous. Moreover, all data will be anonymized during the transcription process, making it impossible to trace transcribed statements to the individual who said them. Finally, it was written expressly that all data would be saved in a password-protected folder, only accessible to the research team. We have, however, rewritten this last section and made it clearer (please read 

“However, as previously mentioned, the main problem of this manuscript is that the authors do not present any results that can be reviewed or contrasted. They mentioned in line 232 a potential limitation due to an over-representation of groups that the reviewers cannot judge because there is no data provided to support any discussion or conclusion.”

-We appreciate reviewer #2’s criticism. However, this manuscript is a study protocol, so it is impossible to present any results by definition. Moreover, the group representation limitations mentioned are predicted limitations inherent to the methodology; we discuss them in an effort to guarantee transparency and guide future conclusions. 

Reviewer #3:

(…) “The major concern is a methodological issue that should be better clarified. It is not clear how you have defined the criteria to "balance" chronic non oncological and oncological patients and healthcare providers. This is really important as the two different populations of patients and also of caregiver providers could have different needs.”

-We thank reviewer #3’s insightful criticism. Although we understand the importance of representativity of different patient groups, we aim to develop a general help line and do not want to artificially influence our sample by ensuring some groups are present in a certain amount. In fact, this would add significant obstacles to our methodology, as we would have to ensure other patient groups are also represented, such as patients with neuropathic, post-surgical or degenerative pain, for example, so that biases are minimised. The target population of the help line is the whole chronic pain population. Perhaps in future studies, when the overall project is in a more advance phase, it would be valuable to obtain perspectives from specific groups within that population, to ensure their particular needs are met. However, as a first step of a long project, we would rather have general sample groups.

“Another concern is always related to the selection of your population who will evaluate the "support line". In the text it is not described if the authors have taken in consideration how to avoid bias related to select people only from countryside/city, type of pain, etc etc. Please describe better which are all the parameters that you consider in the evolution of patients and healthcare providers selected. If you have not planned it please describe it better in limitations of the protocol.”

-We very much appreciate reviewer #3’s constructive criticism. Regarding chronic pain types, as mentioned in the previous reply, we deliberately chose not to differentiate them in this phase, and will contact a general chronic pain patient association. As for the population limitations, we accept this as a methodology by-product. We have further explored this issue in the discussion, and believe it is now more clear and detailed (please read the changes on the second paragraph of the discussion section). 

“The authors should describe better the protocol in order to permit the possibility to replicate in another country if possible.”

-We thank the reviewer’s remarks and have extensively rewritten the Materials and Methods section, making it clearer and more detailed.

---

## [Decision Letter · Decision Letter 1]

5 Aug 2022

Patients and healthcare professionals perspectives on creating a chronic pain support line in Portugal: a qualitative study protocol.

PONE-D-21-35572R1

Dear Dr. Cruz,

We’re pleased to inform you that your manuscript has been judged scientifically suitable for publication and will be formally accepted for publication once it meets all outstanding technical requirements.

Kind regards,

Sara Rubinelli

Academic Editor

PLOS ONE

Additional Editor Comments (optional):

Reviewers' comments:

Reviewer's Responses to Questions

**Comments to the Author**

1. Does the manuscript provide a valid rationale for the proposed study, with clearly identified and justified research questions?

Reviewer #3: Yes

2. Is the protocol technically sound and planned in a manner that will lead to a meaningful outcome and allow testing the stated hypotheses?

Reviewer #3: Yes

3. Is the methodology feasible and described in sufficient detail to allow the work to be replicable?

Reviewer #3: Yes

4. Have the authors described where all data underlying the findings will be made available when the study is complete?

Reviewer #3: No

5. Is the manuscript presented in an intelligible fashion and written in standard English?

Reviewer #3: Yes

6. Review Comments to the Author

You may also provide optional suggestions and comments to authors that they might find helpful in planning their study.

Reviewer #3: The authors have satisfactory replied to all concerns previously raised. The text is now clearer and more readable

7. PLOS authors have the option to publish the peer review history of their article (what does this mean?). If published, this will include your full peer review and any attached files.

Reviewer #3: No

---

## [Editor Report · Acceptance letter]

9 Aug 2022

PONE-D-21-35572R1 

Patients and healthcare professionals perspectives on creating a chronic pain support line in Portugal: a qualitative study protocol 

Dear Dr. Cruz:

I'm pleased to inform you that your manuscript has been deemed suitable for publication in PLOS ONE. Congratulations! Your manuscript is now with our production department. 

Kind regards, 

on behalf of

Dr. Sara Rubinelli 

Academic Editor

PLOS ONE